# Water content for clot composition prediction in acute ischemic stroke

**Kenichi Sakuta**[1,2,3], **Taichiro Imahori**[1,4], **Amir Molaie**[3], **Mahsa Ghovvati**[1], **Neal Rao**[3], **Satoshi Tateshima**[1], **Naoki Kaneko**[1] *

**1** Department of Radiological Sciences, David Geffen School of Medicine at UCLA, Los Angeles, CA, United States of America, **2** Department of Neurology, Jikei University School of Medicine, Tokyo, Japan, **3** Department of Neurology, David Geffen School of Medicine at UCLA, Los Angeles, CA, United States of America, **4** Department of Neurosurgery, Kitaharima Medical Center, Hyogo, Japan

* nkaneko@mednet.ucla.edu

## Abstract

### Background

Mechanical thrombectomy (MT) has become the gold standard care for treating acute ischemic stroke (AIS) due to large vessel occlusion. Emerging evidence suggests that understanding the composition of clots prior to intervention could be useful for the selection of neuroendovascular techniques, potentially improving the efficacy of treatments. However, current imaging modalities lack the ability to distinguish clot composition accurately and reliably. Since water content can influence signal intensity on CT and MRI scans, its assessment may provide indirect clues about clot composition. This study aimed to elucidate the correlation between water content and clot composition using human clots retrieved from stroke patients and experimentally generated ovine clots.

### Materials and methods

This study involved an analysis of ten clots retrieved from patients with AIS undergoing MT. Additionally, we created ten red blood cells (RBC)-rich and ten fibrin-rich ovine blood clots, which were placed in a human intracranial vascular model under realistic flow conditions. The water content and compositions of these clots were evaluated, and linear regression analyses were performed to determine the relationship between clot composition and water content.

### Results

The regression analysis in human stroke clots revealed a significant negative association between RBC concentration and water content. We also observed a positive correlation between water content and both fibrin and platelets in ovine blood clots.

### Conclusion

We identified a significant inverse relationship between clot RBC concentration and water content. Accurate detection of this feature through diagnostic imaging could be beneficial for preoperative clot characterization and planning in MT for AIS.

**Editor:** Sonu Bhaskar, Global Health Neurology Lab / NSW Brain Clot Bank, NSW Health Pathology / Liverpool Hospital and South West Sydney Local Health District / Neurovascular Imaging Lab, Clinical Sciences Stream, Ingham Institute, AUSTRALIA

**Data Availability Statement:** The dataset relevant to this study cannot be shared publicly due to the presence of potentially identifying patient data. We confirm that all non-identifying data underlying the findings are fully available without restriction. All relevant non-identifying data are included within the paper.

**Funding:** This study was funded by Society of Vascular and Interventional Neurology (SVIN) Pilot Grant to IT and NK. The equipment and material purchase were supported by Tarsadia Foundation, Jennifer Carroll Wilson Aneurysm Foundation and Frederick Gardner Cottrell Foundation. The funders had no role in study design, data collection and analysis, decision to publish, or preparation of the manuscript.

**Competing interests:** The authors have declared that no competing interests exist.

## Introduction

Acute ischemic stroke (AIS) is a major cause of fatality and disability around the world [1]. A significant cause of cerebral infarction is large vessel occlusion (LVO), a condition in which major brain arteries are blocked, often leading to severe ischemic strokes. LVOs are associated with high rates of disability and mortality, making them a critical focus in stroke research and treatment [2,3]. Mechanical thrombectomy (MT), a procedure to remove the occluding clot, has become a gold standard treatment for ischemic stroke caused by LVO [4,5]. The most common MT methods involve the use of an aspiration catheter, stent retriever, or a combination of both [6]. While these techniques have demonstrated high rates of successful recanalization [7], achieving successful reperfusion with a single pass–deemed the "first-pass effect"–is associated with improved clinical outcomes and fewer procedure-related complications [8–10].

Clot pathologies play a crucial role in influencing the success of MT and achieving first-pass effect [11,12]. The stent retriever technique, for instance, demonstrates greater efficacy in removing red blood cells (RBC)-rich clots compared to fibrin-rich clots [13–15]. However, very soft RBC clots can be fragmented by the complete opening of the stent retriever [16]. On the other hand, fibrin-rich pathology can reduce the efficacy of stent retrievers, as fibrin-rich clots are not easily penetrated by the stent struts. In contrast, aspiration has demonstrated efficacy in removing very soft clots [16], although it may encounter difficulties with large, delicate clots or moderately stiff ones. A combined approach, harnessing- both an aspiration catheter and stent retriever- might offer a synergistic advantage, potentially elevating treatment efficacy. An in-vitro study showed that the combined technique achieved the highest success rates in removing stiff, calcified clots [17]. Consequently, determining clot pathology prior to MT could be useful for selecting the most effective initial treatment strategy [18–21].

Currently, there are no diagnostic imaging techniques that can reliably and consistently detail clot pathologies. Notably, the process of clot formation within blood vessels is complex and influenced by numerous factors, such as blood composition, the condition of the vessel wall, and blood flow [22,23]. Additionally, clots exhibit heterogeneous pathologies, with varying proportions of fibrin, RBCs, platelets, and white blood cells (WBC) as their primary compositions [24]. The hyperdense sign on CT or the blooming artifact observable on Susceptibility-Weighted Imaging (SWI) on MRI are useful for identifying RBC-rich clots [15,25,26]. However, there are no diagnostic imaging techniques that can elucidate clot characteristics reliably. Conversely, without predominant composition of RBCs, it is challenging to ascertain the extent of RBCs, fibrin, and platelets that make up the pathology from preoperative images. Recent studies have indicated that the complex behavior of animal clots correlates with their water content, and that differences in water content in animal in vitro clots might be detectable via CT [27,28]. Nonetheless, despite these findings, the relationship between water content and the pathology of human clots remains unknown.

In this study, we aimed to bridge this gap by examining the relationship between clot composition and water content in human clots harvested from patients with LVO, validating our findings in artificial animal blood clots.

## Methods

### Human clots from stroke patients

The study involving human blood clot collection was reviewed and approved by the Institutional Review Board (IRB) and the requirement for informed consent was waived for this study. We analyzed blood clots retrieved from AIS patients who underwent MT for LVO between July 2022 through February 2023. Indications for MT were determined according to

the American Heart Association / American Stroke Association guidelines, and the procedure was performed after informed consent was obtained from each patient or the patient's family [5]. The devices used were FDA-approved and commonly available devices, and the treatment strategy was determined by the neuroendovascular physicians [6].

The clots were divided into two sections: one portion was used to measure water content, while the other portion was utilized for pathology slides. For the pathological analysis, the slides were stained with Martius Scarlet Blue (MSB), and an image analysis software was employed to calculate the percentages of various clot components. The correlation between the clot constituents and their water content was then statistically analyzed.

## Ovine blood clot preparation

Ten RBC-rich clots and ten fibrin-rich clots were prepared using the modified Chandler loop method to create realistic clot analogs using ovine blood with sodium citrate from HemoStat Laboratories [29,30]. Polyvinyl chloride tubing (Length 31 cm and diameter 8 mm) was used for the loop and filled with the anticoagulated blood (9mL blood and 0.9mL calcium chloride). For RBC-rich clots, whole blood in the loops was then recalcified to reverse the anticoagulant effects by the addition of 10% calcium chloride. For fibrin-rich clots, plasma was prepared by centrifuging the whole blood carefully at low speed (550 g for 15 minutes at room temperature) and the fibrin-rich clot mixture was constituted by combining 99% of the plasma with 1% RBCs and the calcium chloride. The tubing, filled with either whole blood or plasma, was placed in the Chandler Loop apparatus and incubated at 37˚C with a constant rotation speed of 10 rpm for 1 hour. Following incubation, the clots were carefully retrieved from the loops, ensuring minimal disruption to their structure. They were then stored in phosphate-buffered saline (PBS) at a temperature of 4˚C until ready for use.

## In-vitro thrombectomy model

The in-vitro thrombectomy experimental system was constructed in accordance with the previously described protocol [31]. A pulsatile pump (Harvard Apparatus, MA, US) was connected to physiological flow loop and a silicone replica of human neurovascular anatomy, consisting of the internal carotid artery (ICA), posterior communicating artery, anterior communicating artery, and middle cerebral artery (MCA). PBS was circulated within the experimental system with the pulse rate at a 60 beats per minute (bpm), a flow rate of 240 mL/min for the ICA, a pressure of 120/80 mmHg, and a maintained temperature at 37˚C. To induce occlusion in the MCA M1 segment, the clot was trimmed to 10 mm for RBC-rich clots and 7 mm for fibrin-rich clots. The clot analogs were then inserted into the cervical ICA and naturally positioned themselves within the M1 segment of the MCA under dynamic flow conditions. The clots were left to settle for 3 minutes to ensure stable and realistic occlusion. Following this settling period, the clot was extracted from the model for MSB staining by reversing the fluid flow in the system.

## Pathological analysis

The clots were fixed in a 4% paraformaldehyde solution for 24 hours and then longitudinally embedded in paraffin blocks. The clots were sliced into sections with a thickness of 4 μm. Selected sections, two per clot, were stained with a commercially available MSB kit (MSB Stain Kit RS4607-500, AVANTIK, US). High-resolution images of the MSB-stained clot slides were obtained using a digital slide scanner. These images were analyzed with Orbit Image Analysis software (www.orbit.bio), which utilizes machine learning algorithms to differentiate clot components, including RBC, WBC, fibrin, and platelet/others. The software performed

segmentation of the clot images into these components and subsequent analysis calculating the percentage of each clot component.

## Water content measurement

The weight of the collected clots was measured with an analytical balance (METTLER TOLEDO, USA). Then, the clot was dried at 90˚C for 8 hours to allow the water to evaporate sufficiently. The water content % was calculated using the formula: [(Initial wet weight–Dry weight)/Initial wet weight] ×100%, thus determining the proportion of water in the clot.

## Statistical analyses

For the analysis of the relationship between water content and clot composition, a linear regression analysis was conducted. The linearity of the relationship was assessed through scatter plots and the residuals were checked to ensure that assumptions of homoscedasticity and normality were met. The statistical model was as follows:

Water content = β0 + β1*(Composition of clot)

In the model, β0 represents the y-intercept (the estimated water content when the composition of the clot is zero), β1 is the regression coefficient (the change in water content for each unit change in the composition of the clot). The significance of the regression model was evaluated using an F-test, and the strength and direction of the relationship were determined by the coefficient of determination ($R^2$) and the sign of the β1 coefficient, respectively. The regression coefficient (β1) was reported with a 95% confidence interval to provide an estimate of the uncertainty around the prediction. Student t-test was used for the comparison of water content between RBC-rich and fibrin-rich clots.

All data are presented as mean ± standard deviation. A p-value of <0.05 was considered statistically significant. All statistical analyses were performed using SPSS (v23 for Windows; SPSS Inc., Chicago, IL, USA) statistical software package.

## Results

### Human stroke clots

In this study, we analyzed human stroke clots from 10 patients with AIS who underwent MT for LVO, as shown in Table 1. The mean age of patients was 77 years, and 8 were male. Seven in ten patients had strokes due to cardioembolic etiology, while two in ten had strokes considered cancer-related (patient No.6 and No.10). One patient (No.5) had large artery atherosclerosis and was concurrently diagnosed with COVID-19. Intravenous thrombolysis with tissue plasminogen activator was administered to 7 patients. The average number of passes required for successful thrombectomy was 2.4.

**Fig 1** demonstrates the heterogeneity in primary clot compositions among the patient cohort, arranged by ascending water content. On average, human stroke clots was consisted of 30.04% RBCs, 7.50% white blood cells, 36.43% platelets, and 26.04% fibrin. In patients with cancer-related strokes (patient No.6 and No.10), platelets were the predominant component of the clot material, constituting 54% and 75%, respectively. In contrast, cellular elements, including red and white blood cells, were the least, accounting for 21% and 3%, respectively. The results of the regression analysis are shown in **Table 2**, which reveals an inverse association between the proportion of RBCs and water content—a lower percentage of RBCs was significantly correlated with increased water content in the clots. Pearson correlation analysis further indicated a statistically significant relationship between the fibrin and platelets fraction in human clots and water content, with correlation coefficient of r = 0.643 (p = 0.045).

**Table 1. Breakdown of the patients underwent thrombectomy.**

| Patient No. | age | sex | past medical history | NIHSS | thrombolysis | occluded vessel | TOAST | thrombectomy technique | number of device pass | eTICI |
|---|---|---|---|---|---|---|---|---|---|---|
| 1 | 46 | Male | AF | 16 | 1 | Right MCA | Cardioembolic | combined | 5 | 2b |
| 2 | 74 | Male | HT, HL, AF | 18 | 0 | Left MCA | Cardioembolic | stent retriever | 1 | 3 |
| 3 | 68 | Male | AF | 10 | 1 | Left MCA | Cardioembolic | combined | 1 | 3 |
| 4 | 96 | Male | CAD | 18 | 1 | Right MCA | Cardioembolic | aspiration | 2 | 3 |
| 5 | 84 | Male | COVID-19 | 18 | 1 | Left ICA | Large artery atherosclerosis | aspiration | 3 | 2c |
| 6 | 78 | Male | HT, HL, AF, CAD, CKD pancreatic cancer | 22 | 0 | Left ICA | Other (hypercoagulable state) | combined | 6 | 2a |
| 7 | 95 | Female | HT, HL, AF | 24 | 1 | Left MCA | Cardioembolic | combined | 1 | 2c |
| 8 | 86 | Male | HT, HL, CAD | 23 | 1 | Basilar artery | Cardioembolic | combined | 1 | 3 |
| 9 | 58 | Male | HT | 5 | 1 | Left ICA | Cardioembolic | combined | 2 | 2b |
| 10 | 84 | Female | HT,HL,DM adenocarcinoma | 4 | 0 | Left MCA | Other (hypercoagulable state) | combined | 2 | 3 |

Abbreviations: AF, Atrial Fibrillation; CAD, Coronary Artery Disease; CKD, Chronic Kidney Disease; COVID, Coronavirus Disease 2019; HL, Hyperlipidemia; HT, Hypertension; ICA, Internal Carotid Artery; MCA, Middle cerebral artery; MI, Myocardial Infarction; NIHSS, National Institutes of Health Stroke Scale; eTICI, extended Thrombolysis in Cerebral Infarction; TOAST, Trial of Org 10172 in. Acute Stroke Treatment.

### Ovine blood clots

Fig 2 shows the composition and water content of ovine blood clots. Clots labeled 1 to 10 were formed as RBC-rich clots, whereas clots labeled 11 to 20 were created as fibrin-rich clots. The pathological analysis revealed that RBC-rich ovine clots presented with a notably higher proportion of RBCs at 60.50%, accompanied by 7.91% WBCs, 15.05% PLTs, and 16.55% fibrin on

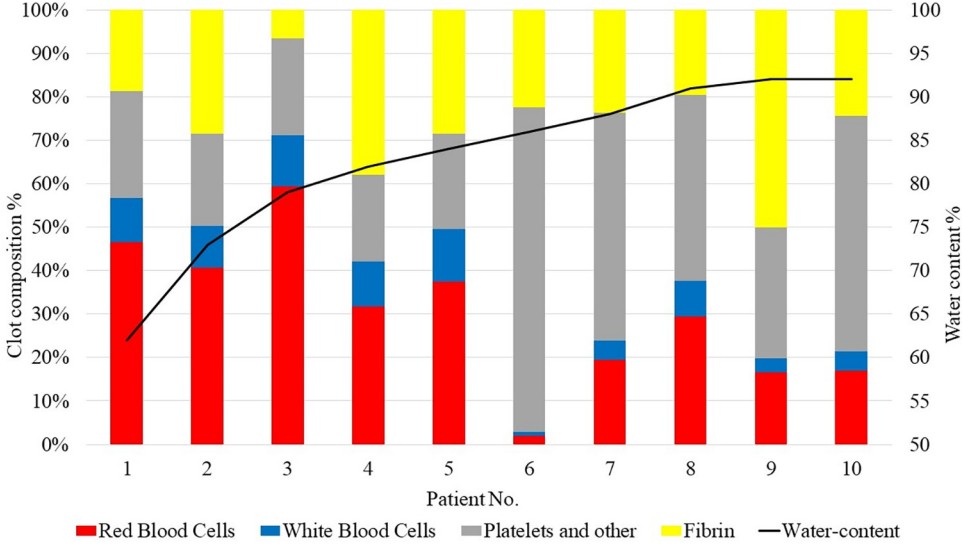

**Fig 1. The clot composition and water content of human stroke clots.** A bar graph illustrates the histological composition of human stroke clots, determined using Martius Scarlet Blue staining, with the following color representations: Red for red blood cell, blue for white blood cell, yellow for fibrin, and grey for platelet and others. The water content of each human stroke clot is concurrently shown in a line graph.

**Table 2. Results of the regression analysis in human clots.**

|  | β0 | β1 | P value | R² |
|---|---|---|---|---|
| Fibrin | 76.049 | 26.309 | 0.364 | 0.104 |
| Platelets and other | 73.606 | 25.514 | 0.139 | 0.253 |
| White Blood Cells | 92.876 | -133.084 | 0.098 | 0.305 |
| Red Blood Cells | 93.998 | -36.944 | 0.040 | 0.427 |

average. On the other hand, the fibrin-rich ovine clots demonstrated significantly lower RBC content at 8.68%, WBCs at 4.09%, a substantial amount of PLTs at 45.86% and fibrin at 41.37%. The average water content was significantly lower in the RBC-rich clots than in fibrin-rich clots, with values of 84.1±1.0% and 93.6±1.1%, respectively (P<0.001). **Table 3** summarizes the outcomes of the regression analysis, which demonstrated statistical significance of the regression model for all clot components, with the percentage of RBCs exhibiting the strongest predictive value. Similarly, Pearson correlation analysis revealed a statistically significant association between the fibrin and platelets fraction and water content, with a correlation coefficient r = 0.958 (p<0.001).

## Discussion

In this study, we investigated the relationship between clot composition and water content in both human stroke clots and experimentally created ovine blood clots, demonstrating an inverse association between clot water content and RBC percentage. Furthermore, we observed a positive correlation between water content and both fibrin and platelet proportions in animal clots.

Our results are consistent with prior studies demonstrating an inverse relationship between water content and RBC composition in artificially created ovine blood clots [27]. Previously, it was shown that for every 1% increase in RBCs, there was a corresponding 0.26% reduction in

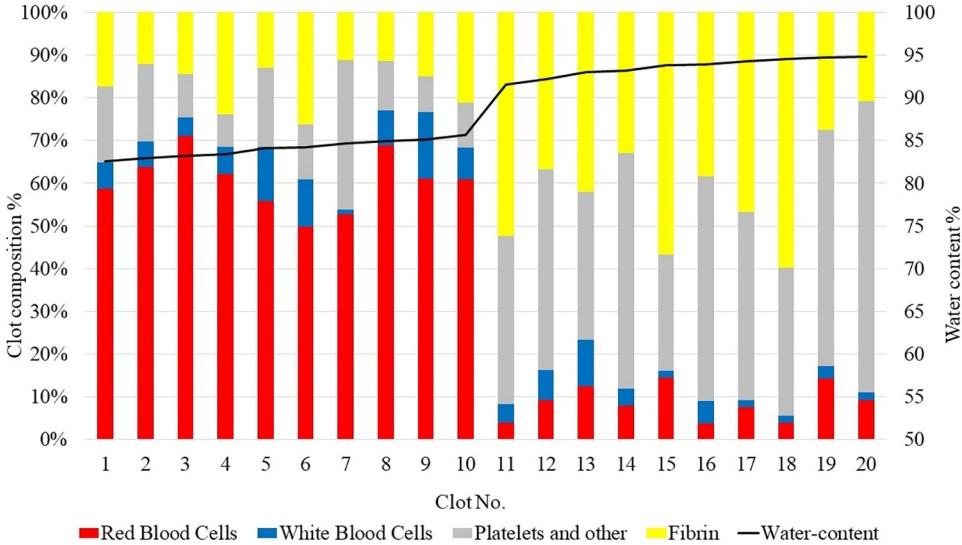

**Fig 2. The clot composition and water content of ovine blood clots.** A bar graph illustrates the histological composition of ovine blood clots, determined using Martius Scarlet Blue staining, with the following color representations: Red for red blood cell, blue for white blood cell, yellow for fibrin, and grey for platelet and others. The water content of each ovine blood clot is concurrently shown in a line graph.

**Table 3. Results of the regression analysis in animal clots.**

| | β0 | β1 | P value | R$^2$ |
|---|---|---|---|---|
| Fibrin | 81.901 | 23.986 | <0.001 | 0.585 |
| Platelets and other | 82.041 | 22.349 | <0.001 | 0.711 |
| White Blood Cells | 92.409 | -59.407 | 0.031 | 0.234 |
| Red Blood Cells | 94.943 | -17.623 | <0.001 | 0.921 |

water content. In the present study, an increase of 1% in RBCs resulted in a 0.37% decrease in human stroke clot water content and a 0.18% decrease in ovine blood clot water content. A potential reason for this inverse relationship between water content and RBC composition could be that RBC may contain a lower water content than other blood components, as detailed below.

Our study also identified a moderate positive correlation between water content and the presence of fibrin and platelets in line with the report by Gonzalez et al [27]. The fibrin network, comprising branched fibers, contributes to water retention due to its mesh-like protein structure [32,33]. However, while ovine blood clots showed a significant correlation between water content and fibrin component, this correlation was not significant in human stroke clots. Brown et al. demonstrated that stretching a clot modifies the arrangement of its mesh structure, leading to a decrease in water content and a reduction in volume [33]. Additionally, Ghezelbash et al. investigated the poroelastic property of a blood clot and identified that under deformation, water migrates in the porous medium of a blood clot [28]. These suggest that the water content in a blood clot is dynamic and can change as the clot deforms due to the movement of water within the clot's structure. Given these insights, the limited correlations in our study may be due to the intrinsic properties and mechanics of these components within the clot.

Overall, our findings contribute to the understanding of the relationship between clot pathologies and their water content. A deeper insight into the water content of clots offers the potential to leverage this property in diagnostic imaging for the identification of specific clot compositions. The ability to predict clot features from preprocedural images could allow neuroendovascular physicians to select the optimal devices, increasing the chances of achieving first-pass success, and improving patient outcomes. There are several avenues of research ongoing to identify methods to detect clot characteristics preoperatively, including the use of machine-learning radiomics [34]. Based on our data, future research should prioritize utilizing readily available diagnostic imaging techniques, such as MRI, to potentially visualize the magnitude of water content within clots.

There are some limitations to this study. First, clots retrieved by MT are subjected to mechanical manipulation by the MT device, which can alter clot structure and its water content [28]. Secondly, the administration of thrombolysis in some patients may have influenced the pathological analysis due to its fibrinolytic effect. This study is a proof of concept and limited by the small number of cases, which restricted our ability to determine the impacts of imaging characteristics with CT or MRI and thrombolysis effects on clot composition. Such studies would significantly contribute to optimizing treatment strategies and improving patient outcomes in acute ischemic stroke management.

## Conclusion

This study identified RBC content as a primary determinant of clot water content. This observation holds promise for the innovation of advanced MRI and CT imaging aimed at

preoperative clot characterization, which could aid in strategic planning in MT procedures for LVO in AIS.

## Author Contributions

**Conceptualization:** Naoki Kaneko.

**Data curation:** Kenichi Sakuta, Taichiro Imahori, Naoki Kaneko.

**Formal analysis:** Kenichi Sakuta.

**Funding acquisition:** Taichiro Imahori, Naoki Kaneko.

**Investigation:** Kenichi Sakuta, Taichiro Imahori, Mahsa Ghovvati, Naoki Kaneko.

**Methodology:** Taichiro Imahori, Naoki Kaneko.

**Project administration:** Neal Rao, Naoki Kaneko.

**Resources:** Neal Rao, Naoki Kaneko.

**Supervision:** Satoshi Tateshima.

**Validation:** Kenichi Sakuta, Taichiro Imahori, Naoki Kaneko.

**Visualization:** Kenichi Sakuta.

**Writing – original draft:** Kenichi Sakuta, Amir Molaie, Naoki Kaneko.

**Writing – review & editing:** Kenichi Sakuta, Taichiro Imahori, Amir Molaie, Mahsa Ghovvati, Neal Rao, Satoshi Tateshima, Naoki Kaneko.

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
