## [Decision Letter · Decision Letter 0]

27 Feb 2024

PONE-D-23-40661Water Content for Clot Composition Prediction in Acute Ischemic StrokePLOS ONE

Dear Dr. Kaneko,

Thank you for submitting your manuscript to PLOS ONE. After careful consideration, we feel that it has merit but does not fully meet PLOS ONE’s publication criteria as it currently stands. Therefore, we invite you to submit a revised version of the manuscript that addresses the points raised during the review process.

We look forward to receiving your revised manuscript.

Kind regards,

Nishant Kumar Mishra, FRCP PhD MD

Academic Editor

PLOS ONE

Additional Editor Comments:

Please tackle reviewer's comments.

Reviewers' comments:

Reviewer's Responses to Questions

**Comments to the Author**

1. Is the manuscript technically sound, and do the data support the conclusions?

Reviewer #1: Yes

Reviewer #2: Yes

2. Has the statistical analysis been performed appropriately and rigorously? 

Reviewer #1: Yes

Reviewer #2: I Don't Know

3. Have the authors made all data underlying the findings in their manuscript fully available?

Reviewer #1: Yes

Reviewer #2: Yes

4. Is the manuscript presented in an intelligible fashion and written in standard English?

Reviewer #1: Yes

Reviewer #2: Yes

5. Review Comments to the Author

Reviewer #1: the authors have performed a study on water contact in cerebral clots and found a correlation between water content and RBCs.

the authors could maybe provide images pf the clot: CT images as well as microscopy in order to illustrate their findings.

This will without doubt impact treatment based on imaging

Reviewer #2: The aim of this study was to gain insight into the possible quantitative relationship between water content and blood elements in clot composition. Human clots obtained from stroke patients and sheep clots obtained in the laboratory were analysed. Although the sample sizes were rather small, some significant associations were found. Notably, there was an inverse association between water content and red blood cell (RBC) content in human clots. Associations between clot composition and medical image features were not evaluated, limiting the translational impact of the results. In addition, the following points should be addressed.

1.- Abstract. Lines 41-43 contain two redundant sentences. They should be reformulated into a single sentence stating the inverse association between water and RBC content. Line 49 should state the nature of the link between water and RBCs.

2.- Ovine Blood Clot Preparation. Line 114. Which anticoagulant was used?

3.- Table 1 should show the device(s)s used in each case (i.e., stent retriever, aspiration or combined).

4.- Although the number of human clots is small, please try to assess a possible association between water/RBC content and the number of device passes (or first effect occurrence) as a measure of clot resistance to MT. If possible, try to include the effect of previous thrombolysis and/or the MT method (device(s)) in the statistical model.

5.- Table 2. Since fibrin and platelets are functionally related in clot formation, some clot composition studies use fibrin+platelets as a clot fraction. Please consider assessing the possible association between fibrin+platelets fraction and water content.

6.- Ovine Blood Clots. Lines 206-207. Should read “The average water content was significantly lower in RBC-rich clots than in fibrin-rich clots”.

7.- Study limitations. Lines 252-253. See comment in point 4 regarding the inclusion of the effect of prior thrombolysis in the statistical model.

6. PLOS authors have the option to publish the peer review history of their article (what does this mean?). If published, this will include your full peer review and any attached files.

Reviewer #1: **Yes: **Karl olof lovblad

Reviewer #2: **Yes: **Juan B. Salom

---

## [Author Response · Author response to Decision Letter 0]

4 Mar 2024

Point-by-point response to the reviewers

I'd like to extend my gratitude to both the editor and the reviewers for the time and effort spent reviewing our manuscript, as well as for the valuable feedback provided.

Reviewer 1 

the authors have performed a study on water contact in cerebral clots and found a correlation between water content and RBCs. the authors could maybe provide images pf the clot: CT images as well as microscopy in order to illustrate their findings.This will without doubt impact treatment based on imaging

Thank you for your constructive feedback on our manuscript. As you mentioned, the imaging analyses would enhance the impact of our study by providing direct visual evidence of the correlation between CT images and clot pathology.

It is important to note that the present study is a proof of concept with a relatively small sample size of human clots and those pre-thrombectomy images were taken with not only CT but also MRI imaging. Therefore, due to the limited number of cases and the variability in imaging modalities, meaningful statistical analysis of imaging data was not feasible. We have added this in the limitation session in Discussion. We believe that future reports will include these relationship.

Reviewer #2

The aim of this study was to gain insight into the possible quantitative relationship between water content and blood elements in clot composition. Human clots obtained from stroke patients and sheep clots obtained in the laboratory were analysed. Although the sample sizes were rather small, some significant associations were found. Notably, there was an inverse association between water content and red blood cell (RBC) content in human clots. Associations between clot composition and medical image features were not evaluated, limiting the translational impact of the results. In addition, the following points should be addressed.

1.- Abstract. Lines 41-43 contain two redundant sentences. They should be reformulated into a single sentence stating the inverse association between water and RBC content. Line 49 should state the nature of the link between water and RBCs.

Thank you for your insightful feedback. Regarding the observation of redundant sentences in lines 41-43, we have carefully reviewed the parts and now revised these lines.

2.- Ovine Blood Clot Preparation. Line 114. Which anticoagulant was used?

We acknowledge the importance of this detail for the reproducibility and clarity of our methods. Sodium citrate was the anticoagulant and has been amended in Line 111.

3.- Table 1 should show the device(s)s used in each case (i.e., stent retriever, aspiration or combined).

We appreciate your suggestion to include the types of devices used in each case in Table 1. we have revised the Table 1 accordingly.

4.- Although the number of human clots is small, please try to assess a possible association between water/RBC content and the number of device passes (or first effect occurrence) as a measure of clot resistance to MT. If possible, try to include the effect of previous thrombolysis and/or the MT method (device(s)) in the statistical model.

Thank you for your suggestion. We recognize the potential value of such an analysis of clot resistance to MT. However, due to the limited number of human clots available in our proof of concept study, it was not feasible to do this analysis with sufficient statistical power. We have added this in the limitation session in Discussion. We believe that future reports will include these relationship.

5.- Table 2. Since fibrin and platelets are functionally related in clot formation, some clot composition studies use fibrin+platelets as a clot fraction. Please consider assessing the possible association between fibrin+platelets fraction and water content.

In response to your suggestion, we have revised our manuscript to include an analysis of the fibrin+platelets fraction in relation to water content. This addition will strengthen the manuscript by offering a broader perspective on how clot composition is related with water content.

6.- Ovine Blood Clots. Lines 206-207. Should read “The average water content was significantly lower in RBC-rich clots than in fibrin-rich clots”.

We have reviewed the mentioned lines and accordingly, we have revised lines 206.

7.- Study limitations. Lines 252-253. See comment in point 4 regarding the inclusion of the effect of prior thrombolysis in the statistical model.

This study is a proof of concept and limited by the small number of cases, which restricted our ability to determine the impacts of imaging characteristics with CT or MRI and thrombolysis effects on clot composition. Such studies would significantly contribute to optimizing treatment strategies and improving patient outcomes in acute ischemic stroke management. We have added this in the limitation session in Discussion.

---

## [Decision Letter · Decision Letter 1]

10 Apr 2024

PONE-D-23-40661R1Water Content for Clot Composition Prediction in Acute Ischemic StrokePLOS ONE

Dear Dr. Kaneko,

Thank you for submitting your manuscript to PLOS ONE. After careful consideration, we feel that it has merit but does not fully meet PLOS ONE’s publication criteria as it currently stands. Therefore, we invite you to submit a revised version of the manuscript that addresses the points raised during the review process.

**ACADEMIC EDITOR: **Thank you for your submission of the revised manuscript. Following careful review and consideration of the feedback provided by the reviewers, we are pleased to inform you that your manuscript has been conditionally accepted for publication, provided the suggestions put forth by the reviewers are addressed (see below). 

One of the reviewers has suggested an addition regarding the "negative/inverse association between RBC and water contents," both in the results (lines 41-42) and conclusion (line 46). We recommend incorporating this suggestion into your manuscript.

We look forward to receiving your revised manuscript.

Kind regards,

Sonu Bhaskar, MD PhD

Academic Editor

PLOS ONE

Journal Requirements:

Reviewers' comments:

Reviewer's Responses to Questions

**Comments to the Author**

1. If the authors have adequately addressed your comments raised in a previous round of review and you feel that this manuscript is now acceptable for publication, you may indicate that here to bypass the “Comments to the Author” section, enter your conflict of interest statement in the “Confidential to Editor” section, and submit your "Accept" recommendation.

Reviewer #2: (No Response)

Reviewer #3: All comments have been addressed

2. Is the manuscript technically sound, and do the data support the conclusions?

Reviewer #2: Yes

Reviewer #3: Yes

3. Has the statistical analysis been performed appropriately and rigorously? 

Reviewer #2: Yes

Reviewer #3: Yes

4. Have the authors made all data underlying the findings in their manuscript fully available?

Reviewer #2: Yes

Reviewer #3: (No Response)

5. Is the manuscript presented in an intelligible fashion and written in standard English?

Reviewer #2: Yes

Reviewer #3: Yes

6. Review Comments to the Author

Reviewer #2: Abstract. It should be clearly stated that there was a negative/inverse association between RBC and water contents, both in results (lines 41-42) and conclusion (line 46).

Reviewer #3: Comments to the Authors:

Thank you for the opportunity to review this manuscript. The authors addressed most of the comments of the reviewers and revised the manuscript accordingly.

However, I believe that Line 49 in the Abstract was not changed to reflect the nature of the link between water and RBCs as requested by Reviewer #2 (Point 1).

Also, I would like to ask the authors to add a short paragraph that summarizes the average for each clot component including human clots and ovine clot analogs.

7. PLOS authors have the option to publish the peer review history of their article (what does this mean?). If published, this will include your full peer review and any attached files.

Reviewer #2: **Yes: **Juan B. Salom

Reviewer #3: No

---

## [Author Response · Author response to Decision Letter 1]

30 Apr 2024

I'd like to extend my gratitude to both the editor and the reviewers for the time and effort spent reviewing our manuscript, as well as for the valuable feedback provided.

Reviewer #2

Abstract. It should be clearly stated that there was a negative/inverse association between RBC and water contents, both in results (lines 41-42) and conclusion (line 46). 

Thank you for your constructive feedback on our manuscript. We have revised the abstract to explicitly state the negative/inverse association between RBC and water contents in both the results and conclusion sections as suggested.

Reviewer #3

Thank you for the opportunity to review this manuscript. The authors addressed most of the comments of the reviewers and revised the manuscript accordingly.

However, I believe that Line 49 in the Abstract was not changed to reflect the nature of the link between water and RBCs as requested by Reviewer #2 (Point 1).

We appreciate your suggestions. I have edited as described above.

Also, I would like to ask the authors to add a short paragraph that summarizes the average for each clot component including human clots and ovine clot analogs.

We have added the average composition data for each clot component, including human clots and ovine clot analogs, into each part in the results section. This addition provides a comprehensive overview of the clot compositions, thereby strengthening the findings presented in our study.

---

## [Editor Report · Decision Letter 2]

14 May 2024

Water Content for Clot Composition Prediction in Acute Ischemic Stroke

PONE-D-23-40661R2

Dear Dr. Kaneko,

We’re pleased to inform you that your manuscript has been judged scientifically suitable for publication and will be formally accepted for publication once it meets all outstanding technical requirements.

Kind regards,

Sonu Bhaskar, MD PhD

Academic Editor

PLOS ONE
---

## [Editor Report · Acceptance letter]

16 May 2024

PONE-D-23-40661R2 

PLOS ONE

Dear Dr. Kaneko, 

I'm pleased to inform you that your manuscript has been deemed suitable for publication in PLOS ONE. Congratulations! Your manuscript is now being handed over to our production team.

Kind regards, 

on behalf of

Dr. Sonu Bhaskar 

Academic Editor

PLOS ONE